# LEARNING INTERPRETABLE NEURAL DISCRETE REPRESENTATION FOR TIME SERIES CLASSIFICATION

## ABSTRACT

Time series classification is a challenging research field with many real-life applications. Recent advances in deep learning have significantly improved the state of the art: recurrent or convolutional architectures allow automatic extraction of complex discriminating patterns that improve performance. Those approaches suffer from a lack of interpretability: the patterns are mapped into a high dimensional latent vector space, they are not representable in the time domain, and are often even not localizable. In this paper, we present a novel neural convolutional architecture that aims to provide a trade-off between interpretability and effectiveness based on the learning of a dictionary of discrete representations. The proposed model guarantees (1) that a small number of patterns are learned, and they are visualizable and interpretable (2) a shift equivariance property of the model associated with a time-consistency of the representation (3) a linear classifier over a limited number of patterns leading to an explainable decision. To ensure the robustness of the discrete representation, they are learned in an unsupervised process independently of the classification task. This allows further great performances in transfer learning. We present extensive experiments on the UCR benchmark wrt usual baselines. The interpretability of the model is illustrated empirically. The chosen trade-off results obviously in a decrease in performance compared to the state of the art. The performance drop is however limited and very dependent on the application domain. The experiments highlight the efficiency of the model for the transfer learning task, showing the robustness of the representations.

## 1 INTRODUCTION

Over the recent years, Deep Neural Networks (DNNs) have become efficient for Time Series Classification (TSC) tasks (Fawaz et al., 2019; Tang et al., 2021). These models are highly capable of extracting complex discriminative features at different frequencies. In the meantime, representation learning approaches demonstrate good performances for clustering (Ma et al., 2019), few-shot learning classification (Franceschi et al., 2019; Malhotra et al., 2017), missing values imputation or forecasting (Zerveas et al., 2022). However, they lack interpretability. Indeed mapping signals into an abstract continuous space of high dimension in which weights have no meaning prevents any justification of the decision resulting from these architectures. In their review of representation-based learning, Bengio et al. (2013) emphasize the ability to extract *Explanatory Factors* and retain *Temporal Coherence* to build good representations. Current approaches do not meet these criteria. Moreover, most of the time, these opaque models are highly specialized in a single task and cannot be transferred to other problems.

In this paper, we propose to discuss the fundamental properties required to build a transparent, explainable, and potentially transferable time series classification process between tasks. First, concerning temporal aspects, we assume that important information has limited support. Therefore, we try to preserve both the shift equivariance of the extracted patterns and the temporal consistency of the latent representation (Cohen & Welling, 2016; Bengio et al., 2013). Second, to achieve a transparent model interpretable by an expert, we impose that the extracted patterns belong to a dictionary of limited size and that their pre-images in the time domain can be computed (Araujo et al., 2019). Third, we restrict ourselves to a strongly regularized linear classifier to explain the decision at the model-level for each task (in addition to an instance-wise interpretation) (James et al.,

2013). Finally, we want a model that is ready for multi-tasking and transfer, i.e. a model where the representation is learned in an unsupervised way, independently from the decision module.

We present a new convolutional architecture relying on neural discrete representation inspired by Van Den Oord et al. (2017) that meets these criteria. Unsupervised representation learning is based on a discrete hierarchical autoencoder. The discretization is a simple vector quantization, which efficiently removes noise in the manner of matrix factorization approaches (Gray, 1984). The hierarchical aspect represents the best compromise to keep a good expressiveness with atoms modeling phenomena at different frequencies while limiting the size of the pattern dictionary (Razavi et al., 2019). For the classification part, a logistic regression is used on the n-grams formed by the successive detections of the dictionary patterns. The model is penalized by an elasticnet regularization (Zou & Hastie, 2005). We demonstrate the competitiveness of this processing chain on UCR data and compare the results to the state of the art while providing qualitative interpretations of the decisions (Dau et al., 2019). We propose an analysis of the origin of performance losses in some classes of applications to better identify the strengths and weaknesses of the different properties related to the interpretability.

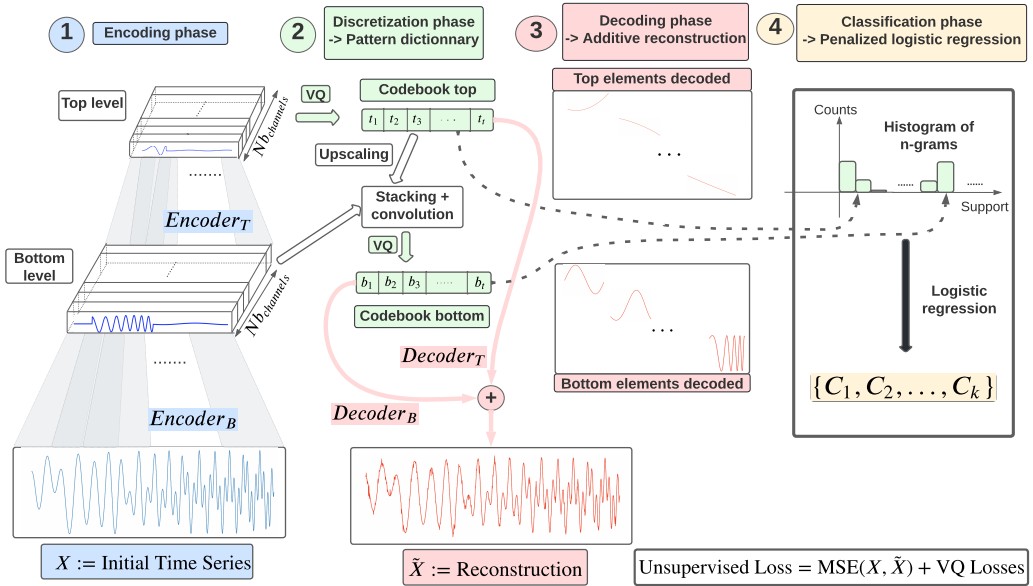

Figure 1: **Overview of the whole process: unsupervised representation learning (1+2+3) and logistic regression on extracted features (4)**. One should notice that the training of the architecture (1+2+3) is independent of the classification task. For more details on encoder's and decoder's structure see Appendix A.1.

**Our main contributions can be summarized as follows:**

- Formalization of the properties required for a more transparent signal classification architecture.
- A model derived from Van Den Oord et al. (2017) taking into account the previous constraints for more interpretability and allowing transfer between tasks.
- A series of experiments demonstrating both the efficiency of the implemented architecture and the qualitative and interpretative interest of the previous constraints.

## 2 INTERPRETABILITY OBJECTIVES FOR TIME SERIES CLASSIFICATION AND GENERALIZATION

In the Time-Series Classification (TSC) field, interpretability can be *instance-wise* or/and *class-wise*. *Instance-wise interpretability* allows weighting the responsible areas of the instance for the

classification decision. But, it does not enable retrieval of class-specific features. *Class-wise interpretability* characterizes a category by a set of features extracted from the representations. Ideally, in the time series framework, these features should be expressive and localizable in time.

K-NN classifiers paired with Euclidean distance (ED) or Dynamical Time Warping (DTW) distance (Rakthanmanon et al., 2012) are popular for time series classification tasks. Although these models are sometimes presented as *instance-wise interpretable* in the sense of reasoning by analogy, they do not allow discriminative feature retrieval for the classification tasks (for both instance and class levels). Most famous *instance-wise interpetability* methods are gradient-based methods. We can mention ResNet Fawaz et al. (2019) that benefits from the *class activation methods* (Zhou et al., 2016) because of the Global Average Pooling layer before the softmax classifier.

Some methods fall between *class-wise interpretability* and *instance-wise interpretability*. The dictionary method developed by Le Nguyen et al. (2019) relies on a linear classifier over a symbolic representation. The extracted symbolic representation is composed of local statistics that could be in the time-domain (Lin et al., 2003) as well as in the frequency-domain (Schäfer & Högqvist, 2012). However, identical elements in the representation can be representative of different patterns. Thus, even if the authors can highlight the areas of interest for each time series, they cannot extract class-specific features.

In this paper, the proposed model aims to build a class-wise interpretable model based on a neural representation of patterns. In the following paragraphs, we list the associated required criteria.

**A temporally consistent representation based on patterns.** One of our main objectives is to build a representation with a meaningful temporal dimension, the elements of which are related to patterns in the initial time series. The time consistency guarantees that the pre-images of two successive elements $a$ and $b$ in the representation are two successive regions ($R_a$ and $R_b$) in the initial time series. These two regions might have a non-empty intersection and their borders are easily computable. Furthermore, an element of the representation must be decodable in the time domain independently of its neighbors. Generally, unsupervised neural models for time series do not allow the interpretation of the extracted features according to the temporal dimension. For example, Franceschi et al. (2019) builds a time series representation using an encoder and a contrastive loss, a triplet loss like Mikolov et al. (2013) where the representation is squeezed along the temporal axis. Secondly, a desirable property for the encoded elements of the representation is the shift equivariance property (Cohen & Welling, 2016). It means that the proposed model should encode two identical patterns with the same value. Moreover, since our model is an autoencoder, we intend two identical elements of the representation to be decoded as the same pattern regardless of their position. Theoretically, it is not easy to satisfy these two properties in an efficient architecture but empirically, we can enforce our model to tend toward this aim.

**A model that can be interpreted by an expert.** By taking all of the above factors into account, we can obtain a time-consistent representation whose elements are identifiable, decodable, and locatable. However, we need additional requirements to make this representation interpretable for a human expert. First, to distinguish the elements of the representation and to make them meaningful, the elements must have discrete and limited support. Van Den Oord et al. (2017); Fortuin et al. (2019) demonstrate that image, audio, or video data can be accurately characterized by discrete factors derived from an autoencoder based on the vector quantization mechanism (Gray, 1984). Secondly, in the time series framework, the temporal dimension of the initial time series has to be compressed through the encoder so that a unique element of the representation must correspond to a (more or less global) pattern that is robust to noise. It is well known that compression is the key to noise robustness, by aggregating similar phenomena (Gillis & Vavasis, 2013). For the compression to be effective and meaningful, several hierarchical levels of representations must be allowed to characterize the different frequencies of the time series (Razavi et al., 2019). The main difference between our proposal and the state of the art lies in the use of a simple additive reconstruction: this approach limits the expressiveness of our model but allows us much greater interpretability of the discrete elements of the representation.

**Multi-tasks & Transfer** The two previous constraints concerned the representation of time series. We impose in our architecture that the classification phase is decoupled from the representation learning, to be able to transfer the encoder-decoder from one application to another. The representation learning is therefore unsupervised, the model being trained exclusively on a reconstruction criterion, in the manner of language models (Devlin et al., 2018). Any classifier can then exploit the time series representation. Franceschi et al. (2019) demonstrated that their trained unsupervised

model could be reused for new time series and that their classifier (SVM with RBF kernel) still performed well on the transferred representation. In contrast, for deep end-to-end models, the gains from transfer seem limited (Fawaz et al., 2018). Our experiments demonstrate the interest of our approach in a transfer setting.

**An explainable decision by a sparse linear model.** The classification decision must preserve interpretability. Linear models such as logistic regression enable the interpretation of the relationship between the extracted features and the class. However, when the logistic regression support is too large, the linear decision is hard to understand. Thus, it is relevant to select the features used for the decision by using $\ell_1$ penalty.

## 3 Model

As explained above, our global objective is 2 folds: build a meaningful, robust & transferable representation of the input time series which respects several fundamental properties on the temporal level on the one hand (localization, translation, ...) and the transparency toward the expert on the other hand. We then train an efficient linear classifier. Our model is depicted in Figure 1.

### 3.1 Unsupervised representation learning

To fulfill the first objective, we choose to use a convolutional auto-encoder architecture with a discretization mechanism. As stated in (Bengio et al., 2013), one of the fundamental principles of representation learning is that the space of hidden factors characterizing the underlying distribution of the data is of a much lower dimension than the space of observations. Besides, dimension reduction using neural networks (Hinton & Salakhutdinov, 2006) allows focusing on important features of the signal while being less sensitive to noise (Van Den Oord et al., 2017). This architecture is built on the neural discrete model of Van Den Oord et al. (2017); Razavi et al. (2019) but we do not learn a probabilistic distribution on the representation space and we use it for interpretability (not for generation purposes). The mechanisms of this architecture allow us to respect the conditions stated in Section 2.

First, at the encoder level, a sequence of convolutions interspersed with non-linearities allows one to learn patterns underlying the time series while respecting the temporal consistency. Furthermore, Cohen & Welling (2016) proves that convolutions with strides of one respect the shift equivariance. If the stride is greater than one, the theoretical shift equivariance is lost (Azulay & Weiss, 2018) because the Nyquist-Shannon sampling theorem (Nyquist, 1928) is not respected anymore. However, using a stride of one does not adequately reduce the dimension. In practice, we prefer to use strides of two and we observe that there is some empirical shift equivariance in that very similar patterns are encoded in the same way despite the dimension reduction and their localization. We have conducted a series of experiments on the *ShapeletSim* dataset (Bagnall et al., 2017a) where we quantify this phenomenon empirically (see Appendix A.2). In the literature, Mouton et al. (2021) has demonstrated that the quality of the empirical shift equivariance depends on the stride (the larger it is the more property is lost) but also on the variance between intermediary elements in a given window. They call this concept, local homogeneity. In practice, in our architecture, an encoder is a succession of convolutions that divides the size of the signal by two (stride of two) at each new convolution. These convolutions are interspersed with LearkyRelu activation functions. Once the desired dimension reduction is reached, we use residual blocks (which do not downscale) for better training stability. For more details on an encoder block see Appendix A.1.

Once the time series has been projected into a vector space of a smaller temporal dimension, we use vector quantization (VQ). To describe the vector quantization mechanism in the context of this autoencoder we use the notations of Van Den Oord et al. (2017). Given a set of $K$ centroids $\{e_k \in \mathbb{R}^d, \ k \in 1, ..., K\}$, vector quantization consists in assigning an output point of the encoder $z_e \in \mathbb{R}^d$ to the nearest centroid :

$$z_q \leftarrow e_k \quad \text{where} \quad k = \arg\min_j ||z_e - e_j||_2^2. \tag{1}$$

The benefit of the VQ mechanism, also known as the self-organizing map (Kohonen, 1990), is that it enables the centroids to move as iterations go on. The number of centroids K is fixed before the learning process and is a hyperparameter of the model. Then, $z_q$ the vector of the point to which

$z_e$ has been assigned is given as input to the decoder. In a nutshell, similar patterns are enforced to be encoded as very similar vectors thanks to empirical shift equivariance. And vector quantization mechanism assigns nearby vectors to the same centroid while respecting temporal coherence.

After having performed the VQ, we pass the quantized vectors to the decoder by ensuring that we keep the original temporality. For the choice of the decoder, we use transposed convolutions whose parameters are symmetrical with those of the encoder. The transposed convolutions are chosen because they allow upscaling of the encoded signal while keeping temporal coherence. Moreover, the transposed convolutions are shift equivariant in translations if the element in the representation to decode is bounded by vectors of $0_{\mathbb{R}^C}$. This property was not true in the instance-wise interpretable classifier of Le Nguyen et al. (2019). However, in practice, there are always edge effects, but the reconstruction between the element and the same shifted element is very close (see Appendix A.2). In practice, a decoder block is a succession of transposed convolutions with a stride of 2. The transposed convolutions are interspersed with LeakyRelu activation functions. We note that the upscaling parameters of the decoder are perfectly symmetrical with the downscaling parameters of the encoder. Moreover, we do not use any bias in our transposed convolutions to avoid the creation of possible artifacts. For more details on a decoder block see Appendix A.1.

Finally, similarly to the architecture of Razavi et al. (2019), we set up two levels of representation: a level that characterizes the high-frequency features and a level that characterizes the low-frequency features (see Figure 1). Thus, for the downscaling phase, there will be two successive encoders. A first encoder whose output will be used to characterize the high frequencies (bottom level) and a second one that takes as input the output of the first encoder and whose output will be used to characterize the low frequencies (top-level). As seen in Figure 1, there will be a quantization for each frequency level. One note that before quantizing the high-frequency information, the output of the bottom encoder must be cleansed of the low-frequency information. Finally, there will be two decoders for the two levels of representation. Each decoder will reconstruct with its information the signal in the initial space (the two decoders are not sequential but parallel). The final reconstruction will be the sum of these two decoded signals.

The unsupervised architecture is trained using the loss below, introduced by Van Den Oord et al. (2017). To simplify the notations we present the loss for a single level model and an instance $x_i$ of a time series with T time steps. The transition to the hierarchical framework is not complicated, it is simply a matter of summing up the different losses linked to the vector quantization for the different levels of representations. Let's introduce some notations. The operator $sg$ stands for the stop gradient operator that blocks gradients during the backward computation time, D is the number of time steps in the representation space, C is the number of channels, and K is the number of available centroids for quantization. The encoder is defined as $\phi_\theta$ and the decoder as $\psi_{\theta'}$. Finally, the matrices involved in the model are $\boldsymbol{E} \in \mathbb{R}^{K \times C}$ the set of K vectors that form the embedding space and $\boldsymbol{Z_q^i} \in \mathbb{R}^{D \times C}$ the temporal vectors sequence after quantization for instance $i$.

$$\arg \min_{\theta, \theta', E} ||\boldsymbol{x_i} - \psi_{\theta'}(\boldsymbol{Z_q^i})||_2^2 + ||sg[\phi_\theta(\boldsymbol{x_i})] - \boldsymbol{Z_q^i}||_2^2 + \beta||\phi_\theta(\boldsymbol{x_i}) - sg[\boldsymbol{E}]||_2^2. \tag{2}$$

The first term of the loss refers to the reconstruction ability of the model. In practice, we consider the mean square error pointwise between the real-time series and the decoder output. This part of the loss optimizes the decoder and the encoder. Although the argmin operator is not differentiable (at the vector at the VQ level in eq.1), Van Den Oord et al. (2017) advise passing the gradients of the embedding vectors to the encoder output vectors that have been assigned to them. Thus the reconstruction loss can optimize the encoder.

The second term of the loss refers to the codebook loss, it moves the embedding vectors according to the vectors that have been assigned to them. However, in practice, we use an exponential moving average optimizer instead. That trick is employed by Van Den Oord et al. (2017) and reproduced in several articles (Razavi et al., 2019). It turns out to be empirically stable to train and allows the choose the solver independently for the rest of the loss. In the model implementation, the solver for the other terms of the loss is Adam (Kingma & Ba, 2014).

The last part of the loss is the commitment loss and ensures that the encoder outputs do not land too far from the embedding vectors. It guarantees better training stability. In practice we choose $\beta = 0.25$ as Van Den Oord et al. (2017) and Razavi et al. (2019).

## 3.2 Classifier insights : logistic regression on histograms of n-grams

Similar to dictionary methods for TSC (Lin et al., 2003; Schäfer & Högqvist, 2012), the proposed representation learning process leads to a numerical representation of the time series: rather than the actual embedding vectors, the series are represented by the sequence of the indices of the vectors in the codebook. To perform the classification task in this context, n-grams are generally extracted and used as input to a K-NN classifier or to a linear classifier. For instance, Le Nguyen et al. (2017) proposes an efficient algorithm to find the most relevant n-grams for a logistic regression classifier.

In the following, only unigrams and bigrams are used for the two levels of representations. We show in Section 4 that with these simple statistics on our representations the results are already close to SOTA for unsupervised models. Bigrams provide a great indication of the local dynamics while remaining in a relatively low dimensional space.

To obtain the final representation of a time series $i$, the histogram $\boldsymbol{h_i}$ of unigrams and bigrams is constructed by concatenating the two histograms of the levels of the representation. Logistic regression is used to solve the binary classification problem:

$$\arg\min_{w,c} \frac{1-\rho}{2}\boldsymbol{w^T w} + \rho\|\boldsymbol{w}\|_1 + C\sum_{i=1}^{n} \log\left(\exp\left(-y_i\left(\boldsymbol{h_i^T w} + b\right)\right) + 1\right), \qquad (3)$$

with $C$ the regularization parameter, $\rho$ the trade-off between the $\ell_1$ penalty and the $\ell_2$ penalty and $b$ the bias. Sparsity in interpretable classification models is desirable to more easily understand the decision (Wu et al., 2018; Li et al., 2019). In the case of multiclass classification, the usual One vs All method (Rifkin & Klautau, 2004) is applied to provide better interpretability.

## 4 Experiments and Results

### 4.1 Quantitative results : classification task on UCR Time Series archive

#### 4.1.1 Protocol and hyperparameters search

In the following, for all considered datasets the number of centroids for vector quantization is fixed at 32 for each scale. Experimentally, this hyperparameter is not very sensitive, but too few vectors lead to worse performances as the representation is unable to catch important features, when too many are useless - numerous atoms correspond to very few representations - and can lead to overfitting.

The main hyperparameters for our unsupervised architecture are the number of convolutional layers of the encoder providing the bottom time-scale reduction and the top time-scale reduction. We explore with a grid search a reduction of $\{4, 8, 16, 32, 64, 128\}$ for the bottom scale and of $\{2, 4, 8, 16, 32\}$ for the top scale. Regarding the classification part, we explore various values for the hyperparameters $\rho$ and $C$ of the logistic regression.

The protocol for the choice of the hyperparameters is the following: the unsupervised architecture is trained for the different possible time-scale reductions on the whole dataset; the training dataset is divided into two parts, 70% used to train the classifier and 30% used for validation. The choice of time-scale reduction is made through the results on the validation dataset. The optimal C and $\rho$ hyperparameters are finally selected by cross-validating on the whole training set[1].

Unfortunately, many of the training data sets in the UCR archive are too small (resulting in a validation set of 4 to 10 instances). Validating on such small data sets does not make sense and gives very variable results (Xu & Goodacre, 2018). Thus, we choose to consider only the datasets with a validation set larger than 30 instances. Furthermore, we restrict the considered datasets to time series with enough time steps ($> 80$) and on supervised problems where the number of classes to predict is between 2 and 6. In summary, 39 datasets meeting these conditions are considered in the following experiments. The presented results report the accuracy of the best model on the test set of each dataset.

To evaluate our classification performance, we compare ourselves to several efficient supervised and unsupervised methods. For supervised models, the considered baselines are: the ensemble method

---

[1]Training for a fixed hyperparameter set can range from 2 minutes to 2 hours depending on the dataset and the dimension reduction. For a fixed dataset, we only use a single Nvidia Titan V for the unsupervised training

HIVE-COTE (Lines et al., 2018), the dictionary method BOSS (Schäfer, 2015), the random convolutional kernels method ROCKET (Dempster et al., 2020), the deep learning methods ResNet (Fawaz et al., 2019) and InceptionTime (Fawaz et al., 2020). For unsupervised models, we compare our model to the results of Franceschi et al. (2019) (Triplet Loss + SVM), and 1-NN's on Euclidian distance and DTW distance. Finally, the random method consists in predicting each time the majority class. The reported results of the baselines are from the 2021 updated results of the survey by Bagnall et al. (2017b) and the results of Franceschi et al. (2019).

### 4.1.2 QUANTITATIVE RESULTS

We present the results as a boxplot (see Figure 2) to see how our method compares to existing models.

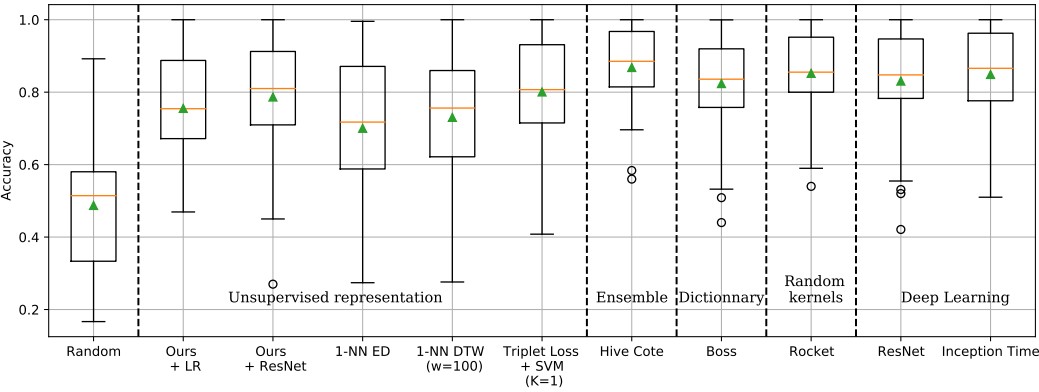

Figure 2: **Boxplot of accuracies**. The results cover the 39 datasets respecting the conditions of the protocol. Triangles represent the arithmetic mean.

The aggregated results presented in the boxplot hide a wide variety of applications: we detail these categories in Table 1 to show that the hierarchy of approaches is variable. Some tasks (like *ChlorineConcentration* or *Yoga*) require patterns learned on discriminative criteria but many data are very well handled by representations from auto-encoders. Other performance drops (on Image and Simulated) are due to the form of the classifier: we can see the gap between logistic regression and ResNet with equal representations. For more details on the results see Appendix A.3.

Table 1: Average Accuracies for unsupervised representation + classifier methods on the 39 datasets of interest by type. The last column shows the results of the best supervised end-to-end model. The best results are in bold and the second best results are underlined (if they exist).

| Type | Count | Ours + LR | Ours + ResNet | Random | 1-NN ED | 1-NN DTW (w=100) | Triplet Loss + SVM (K=1) | Best supervised Hive Cote |
|---|---|---|---|---|---|---|---|---|
| Device | 7 | 0.70 | **0.72** | 0.44 | 0.57 | 0.67 | 0.70 | 0.81 |
| Image | 13 | 0.70 | 0.75 | 0.49 | 0.73 | 0.72 | **0.78** | 0.82 |
| Motion | 4 | 0.91 | 0.91 | 0.53 | 0.86 | 0.84 | **0.93** | 0.95 |
| Sensor | 8 | 0.84 | 0.86 | 0.60 | 0.78 | 0.78 | **0.88** | 0.92 |
| Simu | 1 | 0.78 | **1.00** | 0.26 | 0.91 | **1.00** | **1.00** | 1.00 |
| Spectro | 3 | **0.69** | 0.63 | 0.47 | 0.61 | 0.56 | 0.67 | 0.89 |
| Spectrum | 3 | 0.73 | **0.80** | 0.34 | 0.51 | 0.70 | 0.77 | 0.94 |
| Global accuracy | 39 | 0.76 | 0.79 | 0.49 | 0.70 | 0.73 | **0.80** | 0.87 |

In terms of unsupervised methods, our interpretable model (unsupervised representation + logistic regression) is slightly less accurate than Franceschi et al. (2019). However, when using a powerful

classifier but less interpretable on our representation such as a ResNet (refer to the Appendix A.4 to see the structure of the ResNet in these experiments), the performances are improved. Even in this case, there is still a small gap in accuracy compared to the best supervised models. We believe that this gap is bridged by the following two points. Firstly, our representation can extract features that will subsequently lead to an interpretable and understandable classification (see Section 4.2). Besides, our feature extraction model is reusable for time series of the same size (see Section 4.3).

## 4.2 VISUALIZATION OF INTERPRETABILITY (USECASE: FREEZERREGULARTRAIN DATASET)

To illustrate the interpretability of our model, we focus in this section on the analysis of the results on the FreezerRegularTrain dataset, where it performs well (97% of accuracy). There are two classes, one representing the power demand of the fridge freezer in the kitchen, and the other representing the power demand of the less frequently used freezer in

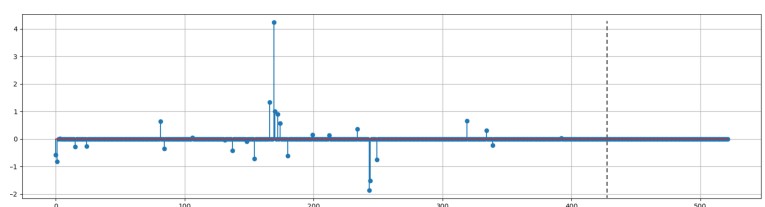

Figure 3: **Coefficients for the logistic regression**. Here are the coefficients of the logistic regression for the FreezerRegularTrain dataset classification problem. The intercept is 0.15. The dotted line represents the boundary between the features from the codebook bottom and features from the codebook top.

the garage. After applying our procedure, we obtain two representations where the bottom codebook length is 19 (downscaling by $2^4$) and the top codebook length is 3 (downscaling by $2^7$). The reconstruction is very accurate with an MSE loss of less than 0.002. Figure 3 shows the weight of each feature of the learned classifier. Only a few features are used as a consequence of the regularization and those features belong mainly to the bottom-scale representation.

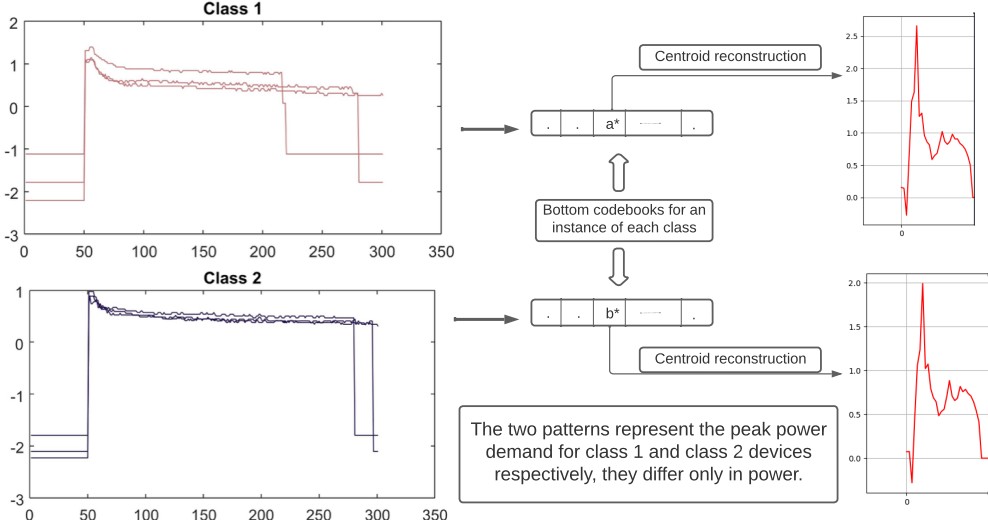

Figure 4: **Visualization of the decoded most important feature for each class according to the logistic regression coefficients**. On the left, is a visualization of a few power curves for each class. On the right, are the most discriminative features for each class. Patterns are rebuilt according to the decoder and individually (in the representation space, neighbor embedding vectors are set to $0_{\mathbb{R}^C}$) Moreover, when we look at the position of the centroid of interest and its receptive field, we notice that both patterns are related to the power demand peak.

Let us note $a^*$ and $b^*$ are the two most discriminative features with the higher coefficients in absolute. Their values are respectively 4.23 and -1.86 (for an intercept of 0.15). Since the logistic regression

acts on histograms of unigrams and bigrams coding for centroid vectors, it is straightforward to retrieve the meaning of $a^*$ and $b^*$. Here, $a^*$ and $b^*$ correspond to unigrams appearing each time at most once in the bottom representation sequence. By reconstructing these centroids through the decoder, we see that these patterns represent the power demand of respectively the class 1 device and the class 2 device (see Figure 4). Thus, without any prior knowledge, it can be inferred from the model that the level of peak power is an important feature to tell apart the two devices. For another interpretable use-case see Appendix A.5.

## 4.3 TRANSFER LEARNING

The other main advantage of our model is to reuse the learned representation for one time series for another time series of the same length to extract a representation. Then, we only have to learn the logistic regression for the specific classification problem and cross-validate it to obtain the adapted hyperparameters ($\rho$ and C). This process is fast and requires only one single CPU for the training phase (for the logistic regression). Table 2 shows the transfer accuracy for datasets where the time series are composed of 80-time steps. For more results, see Appendix A.6. In addition to being fast, the transferred representation enables good accuracy. The results are sometimes even better than the logistic regression on the representation learned on the time series itself. It demonstrates that our unsupervised model can learn and extract key features for different time series without being re-trained. Table 2 shows the transfer accuracy for datasets where the time series are composed of 80-time steps. For more results, see Appendix A.6.

Table 2: Transfer learning on time series of length 80. Columns indicate the source datasets whose unsupervised architectures are re-used. Rows indicate the target datasets. For each target dataset, the target time series is encoded thanks to the unsupervised model learned on the source dataset. The logistic regression is next trained on the obtained representations. Bold accuracies stand for the best results for each target dataset. Underlined accuracies stand for the results when the source and target datasets are the same (without transfer). Dataset names are shortened for the sake of space.

| Target | MTW | MOC | MAG | DAG | DTW | DOC | PPAG | PPTW | PPOC | POC |
|--------|------|------|------|------|------|------|------|------|------|------|
| MTW | 0.49 | 0.53 | 0.52 | 0.47 | 0.49 | 0.47 | **0.57** | 0.47 | 0.54 | 0.52 |
| MOC | 0.70 | 0.70 | **0.74** | 0.65 | 0.69 | 0.70 | 0.70 | 0.66 | 0.70 | **0.74** |
| MAG | 0.55 | 0.56 | 0.49 | 0.57 | 0.57 | 0.55 | **0.62** | 0.53 | 0.55 | 0.55 |
| DAG | 0.63 | 0.71 | 0.74 | 0.68 | 0.73 | 0.70 | 0.70 | 0.71 | 0.65 | **0.77** |
| DTW | 0.58 | 0.58 | **0.63** | 0.57 | 0.57 | 0.53 | 0.61 | 0.58 | 0.53 | 0.58 |
| DOC | 0.71 | 0.74 | **0.75** | 0.69 | 0.69 | 0.75 | **0.75** | 0.67 | 0.67 | **0.75** |
| PPAG | 0.83 | 0.82 | 0.85 | 0.80 | 0.81 | 0.83 | 0.83 | 0.84 | 0.81 | **0.86** |
| PPTW | 0.73 | 0.76 | 0.76 | 0.79 | 0.77 | 0.74 | **0.80** | 0.76 | 0.71 | 0.76 |
| PPOC | **0.82** | 0.78 | 0.80 | 0.78 | 0.80 | 0.79 | 0.77 | 0.74 | **0.82** | 0.80 |
| POC | 0.71 | 0.73 | 0.73 | 0.70 | 0.71 | 0.73 | 0.72 | 0.67 | 0.68 | **0.79** |

For these classification problems which are relatively close, we observe that transferring the unsupervised model and training a logistic regression on top of it allows us to improve the accuracy.

## 5 CONCLUSION AND PERSPECTIVES

The proposed process enables building interpretable representations of time series, classifying accurately for a wide variety of problems, and explaining the classification decision over the representation. The reusable unsupervised model is a step forward toward general semantics of time series as in image or text. For future work, it seems interesting to consider the choice of the reconstruction loss in the unsupervised architecture. Indeed, the mean squared error (MSE) tends to smooth the time series when the model reconstructs them, which can weaken some important features. One can consider using the DILATE Loss (Le Guen & Thome, 2019) instead of the MSE. Another limitation of our process is that we only consider unigrams and bigrams. By reusing the coordinate descent of Ifrim & Wiuf (2011), we can find longer sequences of centroids that allow us to discriminate classes. Finally, the last track to consider is to use autoML methods to automatically find the number of hierarchical levels to use in the unsupervised model.

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

## A APPENDIX

### A.1 INSIDE AN ENCODER BLOCK OR A DECODER BLOCK

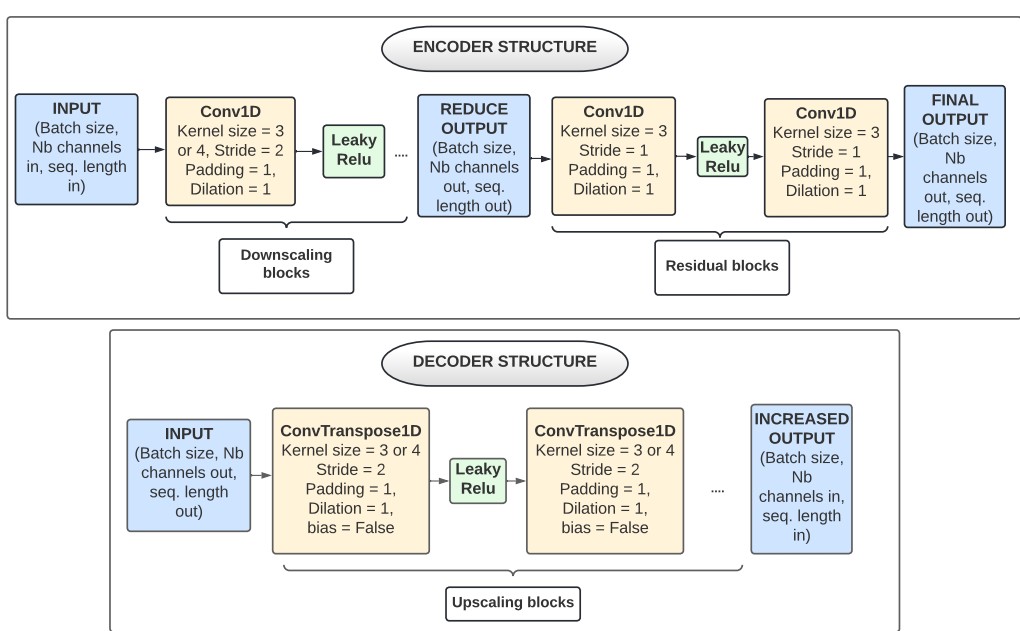

Figure 5: **Encoders and decoders architectures**. The convolution kernels with strides of two allow to contract the temporal dimension and thus to aggregate local information. The dotted line means that several convolution blocks with a stride of two can be added. At the end of the encoding, residual blocks are added to speed up the learning process. In the decoder, the choice of the parameters of the transposed convolutions is perfectly symmetrical with the parameters of the convolutions in the downscaling blocks. The dotted line means that several transposed convolution blocks with a stride of two can be added. There is no bias in these transposed convolutions to avoid potential artifacts.

## A.2 EMPIRICAL SHIFT EQUIVARIANCE, SHAPELETSIM DATASET EXPERIENCE

The theoretical shift equivariance of the non-strided convolutions guarantees that a pattern in the signal which occurs at time t or at time $t + \tau$ is encoded in the same way with a shift $\tau'$. More formally this can be written as follows. Let S be an element of the signal (for instance a subsequence), and let $T$ be the group of discrete translations along the temporal axis. If we take $\tau$ to be any discrete translation in $T$ and f to be a function equivariant by discrete translation for $T$ then there exists $\tau' \in T'$ such that :

$$f(\tau(S)) = \tau'(f(S))$$

In the proposed model, theoretical shift equivariance does not stand because of the strided convolutions (stride of 2). However, we observe that empirically, our model has some empirical shift equivariance in the way patterns are encoded. The experiment below attempts to show this.

In the *ShapeletSim* dataset, there are 100 instances generated with white noise (class 0) and 100 instances generated with white noise as well as a randomly inserted triangle (class 1). The purpose of this experiment is to show that the triangle is encoded (most of the time) in the same way in the codebook regardless of its position in the time series.

After applying our procedure 4.1.1, we obtain two representations where the bottom codebook length is 125 (downscaling by $2^2$) and the top codebook length is 16 (downscaling by $2^5$). The reconstruction loss is 0.12 pointwise. In Figure 6, we see examples of reconstructions for instances of class 1.

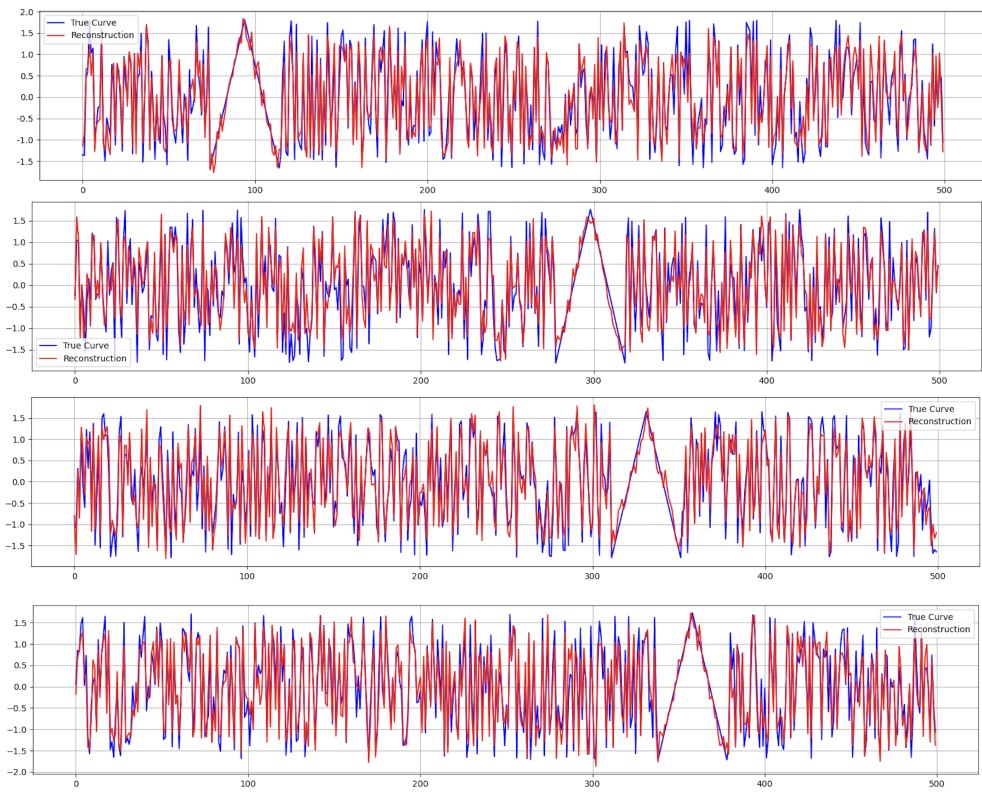

Figure 6: Visualization few time series for *ShapeletSim* dataset where a triangle is encoded at a random position (class 1). The blue curve stands for the true time series and the red one for its reconstruction.

In Figure 7, we see how an instance of class 1 is decomposed between the codebook bottom and the codebook top.

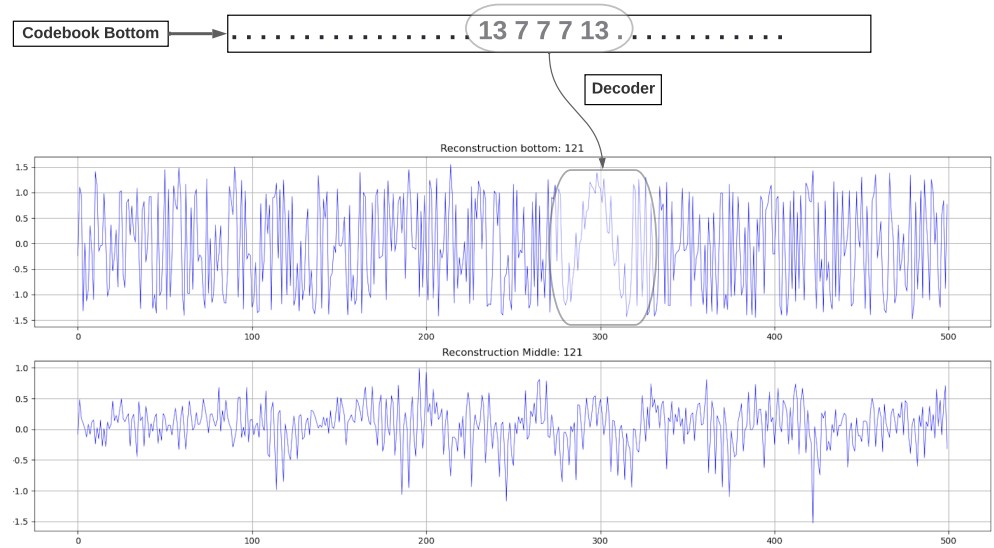

Figure 7: Visualization of how the triangle is encoded for a given position in time according to the bottom/top decomposition.

We notice that for the 100 instances of class 1, the triangles are enconded almost always in the same way. However, there are some edge effects to take into account. Table 3 shows for class 1 the percentage of instances where the triangle is encoded with a given sequence of centroids. We can also see the percentage of instances of class 0 where this given sequence of centroids is present.

Table 3: Pattern of the sub-sequence that encodes the triangle.

| Centroides sequences | White Noise (class 0) | White noise + triangle (class 1) |
|---|---|---|
| '7 7' | 23 % | 100 % |
| '7 7 7' | 1 % | 91 % |
| '13 7 7 7' | 0 % | 88 % |
| '13 7 7 7 13' | 0 % | 75 % |

The Table can be interpreted as follows. In 91 % of the cases of class 1, the triangle is encoded in the bottom codebook by the sub-sequence of centroids '7 7 7'. This sub-sequence of centroids is found in 1 % of the cases of class 0.

To conclude with this experiment, it is interesting to see how simple it is for other algorithms to distinguish class 0 from class 1. We see from ResNet's performance that this task is not so simple.

Table 4: Test accuracy comparison for few methods on the ShapeletSim dataset

|  | Ours + LR | 1-NN DTW (w=100) | 1-NN ED | Triplet Loss (K=1) + SVM | ResNet | Hive Cote |
|---|---|---|---|---|---|---|
| Test Accuracy | 0.95 | 0.65 | 0.54 | 0.89 | 0.73 | 1.0 |

## A.3 ADDITIONAL QUANTITATIVE RESULTS

Table 5: Results and comparisons for unsupervised methods on the 39 datasets of interest. The best results are in bold and the second best results are underlined (if they exist). Sometimes dataset names are shortened for the sake of space.

| Type | Dataset | Ours + LR | Ours + ResNet | Ran-dom | 1-NN ED | 1-NN DTW | Triplet Loss (K=1) + SVM |
|------|---------|-----------|---------------|---------|---------|----------|--------------------------|
| Device | Computers | **0.77** | 0.73 | 0.50 | 0.58 | 0.70 | 0.69 |
| Device | SmallKitchenApp | **0.70** | 0.69 | 0.33 | 0.34 | 0.64 | 0.68 |
| Device | ScreenType | **0.48** | 0.45 | 0.33 | 0.36 | 0.40 | 0.41 |
| Device | RefrigerationDevices | 0.47 | **0.55** | 0.33 | 0.39 | 0.46 | 0.53 |
| Device | ECG200 | 0.77 | 0.84 | 0.64 | **0.88** | 0.77 | 0.87 |
| Device | ECG5000 | 0.92 | 0.93 | 0.58 | 0.92 | 0.92 | **0.94** |
| Device | LargeKitchenApp | 0.77 | **0.83** | 0.33 | 0.49 | 0.79 | 0.80 |
| Image | MiddlePhalanxOC | 0.70 | **0.78** | 0.57 | 0.77 | 0.70 | 0.75 |
| Image | ProximalPhalanxTW | 0.75 | 0.81 | 0.35 | 0.71 | 0.76 | **0.82** |
| Image | ProximalPhalanxOC | 0.82 | 0.85 | 0.68 | 0.81 | 0.78 | **0.87** |
| Image | ProximalPhalanxOAG | 0.83 | 0.84 | 0.49 | 0.79 | 0.80 | **0.85** |
| Image | PhalangesOC | 0.79 | 0.75 | 0.61 | 0.76 | 0.73 | **0.81** |
| Image | MixedShapesRegularTrain | 0.67 | 0.86 | 0.27 | 0.90 | 0.84 | **0.92** |
| Image | MiddlePhalanxTW | 0.50 | 0.56 | 0.29 | 0.51 | 0.51 | **0.60** |
| Image | MiddlePhalanxOAG | 0.49 | **0.63** | 0.57 | 0.52 | 0.50 | 0.62 |
| Image | HandOutlines | 0.88 | 0.89 | 0.64 | 0.86 | 0.88 | **0.92** |
| Image | Yoga | 0.70 | 0.73 | 0.54 | 0.83 | **0.84** | 0.82 |
| Image | DistalPhalanxTW | 0.57 | 0.68 | 0.30 | 0.63 | 0.59 | **0.70** |
| Image | DistalPhalanxOAG | 0.68 | 0.71 | 0.47 | 0.63 | **0.77** | 0.72 |
| Image | DistalPhalanxOC | 0.75 | 0.72 | 0.58 | 0.72 | 0.72 | **0.76** |
| Motion | GunPointOvY | **1.00** | 0.99 | 0.52 | 0.95 | 0.84 | **1.00** |
| Motion | GunPointMvF | 0.97 | 0.99 | 0.53 | 0.97 | **1.00** | **1.00** |
| Motion | GunPointAgeSpan | 0.97 | 0.91 | 0.51 | 0.90 | 0.92 | **0.98** |
| Motion | WormsTwoClass | 0.70 | **0.75** | 0.57 | 0.61 | 0.62 | **0.75** |
| Sensor | FreezerRegularTrain | 0.97 | 0.98 | 0.50 | 0.80 | 0.90 | **0.99** |
| Sensor | Wafer | 0.99 | **1.00** | 0.89 | **1.00** | 0.98 | 0.99 |
| Sensor | StarLightCurves | **0.96** | **0.96** | 0.58 | 0.85 | 0.91 | **0.96** |
| Sensor | ChlorineConcentration | 0.55 | 0.56 | 0.53 | 0.65 | 0.65 | **0.72** |
| Sensor | Earthquakes | 0.73 | **0.75** | **0.75** | 0.71 | 0.72 | **0.75** |
| Sensor | FordA | 0.89 | **0.93** | 0.52 | 0.67 | 0.55 | 0.92 |
| Sensor | PowerCons | 0.91 | 0.94 | 0.50 | 0.93 | 0.88 | **0.96** |
| Sensor | FordB | 0.75 | **0.79** | 0.50 | 0.61 | 0.62 | **0.79** |
| Simu | TwoPatterns | 0.78 | **1.00** | 0.26 | 0.91 | **1.00** | **1.00** |
| Spectro | EthanolLevel | **0.55** | 0.27 | 0.25 | 0.27 | 0.28 | 0.42 |
| Spectro | Strawberry | 0.93 | 0.90 | 0.64 | **0.95** | 0.94 | **0.95** |
| Spectro | Ham | 0.57 | **0.71** | 0.51 | 0.60 | 0.47 | 0.65 |
| Spectrum | SemgHandSubjectCh2 | 0.72 | **0.88** | 0.20 | 0.40 | 0.73 | 0.77 |
| Spectrum | SemgHandGenderCh2 | 0.89 | **0.91** | 0.65 | 0.76 | 0.80 | 0.84 |
| Spectrum | SemgHandMovementCh2 | 0.57 | 0.61 | 0.17 | 0.37 | 0.58 | **0.71** |

Table 6: Results and comparisons for supervised methods on the 39 datasets of interest. The best results are in bold and the second best results are underlined (if they exist). Sometimes dataset names are shortened for the sake of space.

| Type | Dataset | Ours + LR | Ours + Res-Net | Hive Cote | Rock-et | Boss | Res-Net | Inception Time |
|---|---|---|---|---|---|---|---|---|
| Device | Computers | 0.77 | 0.73 | 0.81 | 0.84 | 0.80 | 0.86 | **0.87** |
| Device | SmallKitchenApp | 0.70 | 0.69 | **0.83** | 0.82 | 0.75 | 0.80 | 0.77 |
| Device | ScreenType | 0.48 | 0.45 | 0.72 | 0.61 | 0.58 | **0.76** | 0.71 |
| Device | RefrigerationDevices | 0.47 | 0.55 | **0.56** | 0.54 | 0.44 | 0.52 | 0.51 |
| Device | ECG200 | 0.77 | 0.84 | 0.86 | **0.90** | 0.88 | 0.88 | **0.90** |
| Device | ECG5000 | 0.92 | 0.93 | **0.95** | **0.95** | 0.94 | 0.94 | 0.94 |
| Device | LargeKitchenApp | 0.77 | 0.83 | 0.92 | 0.93 | 0.84 | **0.95** | **0.95** |
| Image | MiddlePhalanxOC | 0.70 | 0.78 | 0.81 | **0.83** | 0.81 | 0.82 | **0.83** |
| Image | ProximalPhalTW | 0.75 | 0.81 | **0.82** | 0.80 | 0.77 | 0.79 | 0.78 |
| Image | ProximalPhalOC | 0.82 | 0.85 | 0.89 | 0.90 | 0.87 | **0.91** | **0.91** |
| Image | ProximalPhalOAG | 0.83 | 0.84 | **0.86** | 0.85 | 0.83 | 0.82 | 0.82 |
| Image | PhalangesOC | 0.79 | 0.75 | 0.83 | 0.84 | 0.82 | 0.85 | **0.86** |
| Image | MixedShapesRegTr | 0.67 | 0.86 | **0.97** | **0.97** | 0.92 | **0.97** | **0.97** |
| Image | MiddlePhalanxTW | 0.50 | 0.56 | 0.58 | **0.59** | 0.53 | 0.53 | 0.53 |
| Image | MiddlePhalanxOAG | 0.49 | 0.63 | 0.70 | **0.71** | 0.66 | 0.60 | 0.59 |
| Image | HandOutlines | 0.88 | 0.89 | 0.92 | 0.94 | 0.90 | 0.94 | **0.95** |
| Image | Yoga | 0.70 | 0.73 | **0.91** | **0.91** | **0.91** | 0.88 | **0.91** |
| Image | DistalPhalanxTW | 0.57 | 0.68 | **0.70** | **0.70** | 0.67 | 0.67 | 0.67 |
| Image | DistalPhalanxOAG | 0.68 | 0.71 | **0.82** | 0.81 | **0.82** | 0.78 | 0.77 |
| Image | DistalPhalanxOC | 0.75 | 0.72 | **0.82** | **0.82** | 0.81 | 0.81 | **0.82** |
| Motion | GunPointOvY | **1.00** | 0.99 | **1.00** | 0.99 | **1.00** | **1.00** | **1.00** |
| Motion | GunPointMvF | 0.97 | 0.99 | **1.00** | **1.00** | **1.00** | 0.99 | **1.00** |
| Motion | GunPointAgeSpan | 0.97 | 0.91 | **1.00** | 0.99 | 0.99 | 0.99 | 0.98 |
| Motion | WormsTwoClass | 0.70 | 0.75 | 0.79 | 0.79 | **0.81** | 0.77 | 0.80 |
| Sensor | FreezerReguTrain | 0.97 | 0.98 | **1.00** | 0.99 | 0.99 | **1.00** | **1.00** |
| Sensor | Wafer | 0.99 | **1.00** | **1.00** | **1.00** | **1.00** | **1.00** | **1.00** |
| Sensor | StarLightCurves | 0.96 | 0.96 | **0.98** | **0.98** | **0.98** | 0.97 | **0.98** |
| Sensor | ChloConcentration | 0.55 | 0.56 | 0.73 | 0.80 | 0.66 | 0.84 | **0.86** |
| Sensor | Earthquakes | 0.73 | **0.75** | **0.75** | **0.75** | **0.75** | 0.72 | 0.73 |
| Sensor | FordA | 0.89 | 0.93 | 0.94 | 0.94 | 0.92 | 0.93 | **0.96** |
| Sensor | PowerCons | 0.91 | 0.94 | **0.99** | 0.96 | 0.89 | 0.89 | **0.99** |
| Sensor | FordB | 0.75 | 0.79 | **0.93** | 0.81 | 0.91 | 0.82 | 0.86 |
| Simu | TwoPatterns | 0.78 | **1.00** | **1.00** | **1.00** | 0.99 | **1.00** | **1.00** |
| Spectro | EthanolLevel | 0.55 | 0.27 | 0.85 | 0.63 | 0.51 | 0.85 | **0.88** |
| Spectro | Strawberry | 0.93 | 0.90 | **0.98** | **0.98** | 0.97 | 0.97 | **0.98** |
| Spectro | Ham | 0.57 | 0.71 | 0.84 | **0.86** | 0.84 | 0.81 | 0.85 |
| Spectrum | SemgHandSCh2 | 0.72 | 0.88 | **0.95** | 0.91 | 0.84 | 0.55 | 0.76 |
| Spectrum | SemgHandGCh2 | 0.89 | 0.91 | **0.97** | 0.92 | 0.89 | 0.81 | 0.88 |
| Spectrum | SemgHandMCh2 | 0.57 | 0.61 | **0.89** | 0.65 | 0.66 | 0.42 | 0.55 |

Table 7: Results for supervised methods on the 39 datasets of interest by type. The best results are in bold and the second best results are underlined (if they exist). Sometimes dataset names are shortened for the sake of space.

| Type | Count | Ours + LR | Ours + ResNet | Hive Cote | Rocket | Boss | ResNet | Inception Time |
|---|---|---|---|---|---|---|---|---|
| Device | 7 | 0.70 | 0.72 | 0.81 | 0.80 | 0.75 | **0.82** | 0.81 |
| Image | 13 | 0.70 | 0.75 | **0.82** | **0.82** | 0.79 | 0.80 | 0.80 |
| Motion | 4 | 0.91 | 0.91 | **0.95** | 0.94 | **0.95** | 0.94 | **0.95** |
| Sensor | 8 | 0.84 | 0.86 | **0.92** | 0.90 | 0.89 | 0.90 | **0.92** |
| Simulated | 1 | 0.78 | **1.00** | **1.00** | **1.00** | 0.99 | **1.00** | **1.00** |
| Spectro | 3 | 0.69 | 0.63 | 0.89 | 0.82 | 0.77 | 0.88 | **0.90** |
| Spectrum | 3 | 0.73 | 0.80 | **0.94** | 0.83 | 0.80 | 0.59 | 0.73 |

## A.4 RESNET ARCHITECTURE ON REPRESENTATION AND TRAINING PROCEDURE

Figure 8: **ResNet architecture on our unsupervised representation**. It is important to note that in this case, the Resnet model takes as input the vectors of the representation and not the indices. This allows the model to take into account the distance between the centroids.

**Procedure to train the ResNet on representation.** For the choice of the representation, we take the same representation as for the logistic regression. Then, we split the initial training data into training (70%) and validation (30%). For this training, we choose the following two hyperparameters:

- weight decay (research in a grid of 15 values between $10^{-6}$ and $10^{-1}$)
- Number of epochs (research in a grid of 10 values between 50 and 500).

The validation dataset allows us to select the right hyperparameters. One notice that no hyperparameters research is done on the architecture itself. The ResNet model aims to show that it is easy to improve the accuracy by using a non-linear model on the representation.

### A.5 Additional results for interpretability : PowerCons dataset

This dataset contains the individual household electric power consumption for one day. The sampling rate is 10 minutes, so the time series is 144 long. Time series are divided into two classes: class 1 stands for the warm season and class 2 for the cold season.

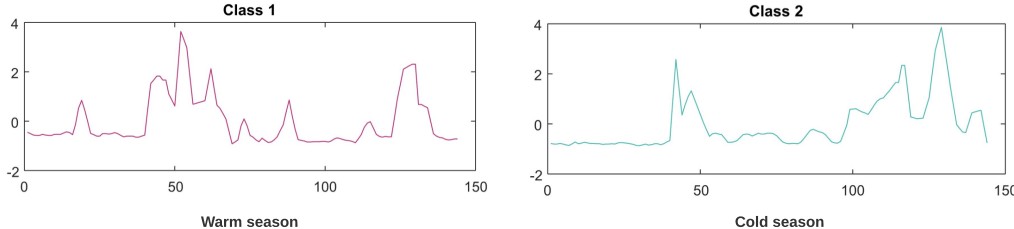

Figure 9: **Overview of an instance for each class.**

After applying our procedure, we obtain two representations where the bottom codebook length is 18 (downscaling by $2^3$) and the top codebook length is 3 (downscaling by $2^3$). The reconstruction loss is around 0.06 pointwise. In Figure 10, we see the decomposition between the different frequency levels.

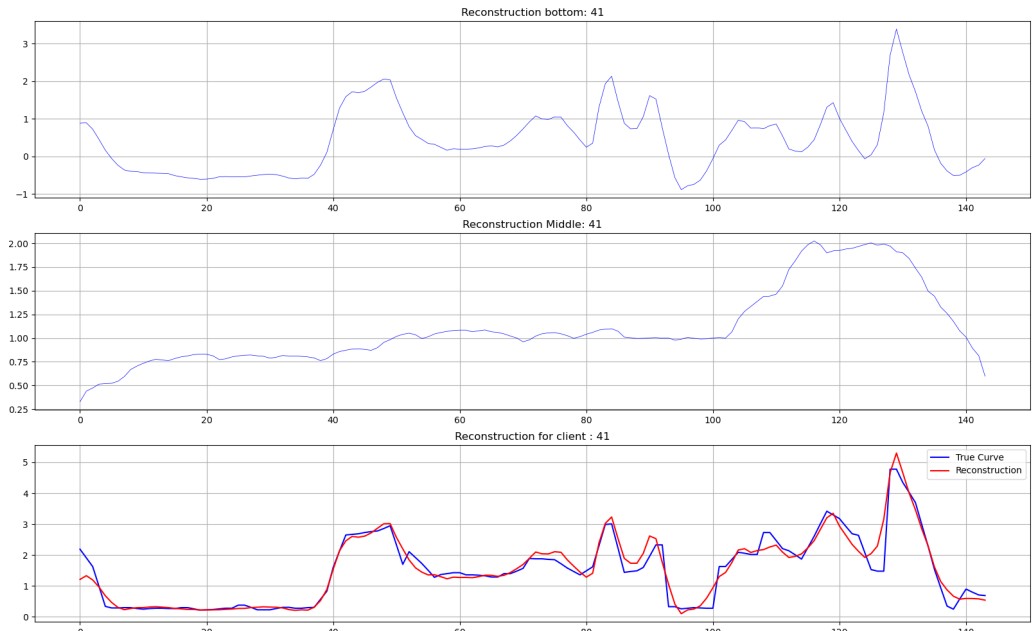

Figure 10: **Bottom/Top decomposition for an instance.** In the bottom plot, the blue curve stands for the initial time series, and the red curve stands for the reconstructed time series.

Once the unigrams and bigrams of the representations are extracted, we train a penalized logistic regression. The Figure 11 shows the coefficients of this logistic regression. Unlike the use-case 4.2, the top-level features seem to be discriminating for the classification problem.

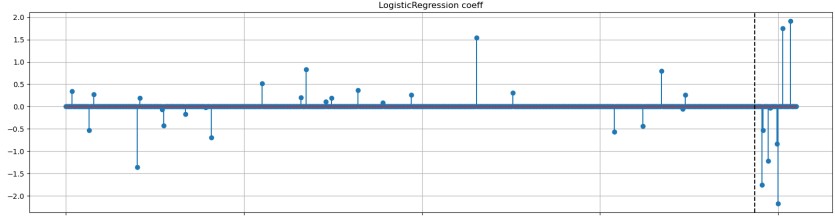

Figure 11: **Logistic regression coefficients**. Intercept is 2.27. Top three features are bigram '9 7', unigram '26' and unigram '1' for the top level, their coefficient are respectively -2.17, 1.91, -1.76. The dotted line represents the boundary between the features from the codebook bottom and features from the codebook top.

Now let's look at the two most discriminating unigrams in detail. They are both extracted from the top codebook.

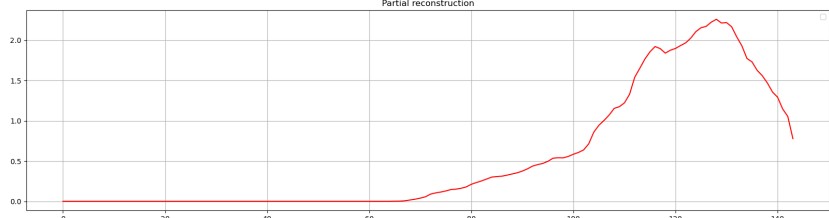

Figure 12: **Decoded unigram '26' at third position** (which is the case most of the time in our setting)

In Figure 12, we see that the pattern favorably characterizing the 'cold season' class shows a high power consumption in the evening.

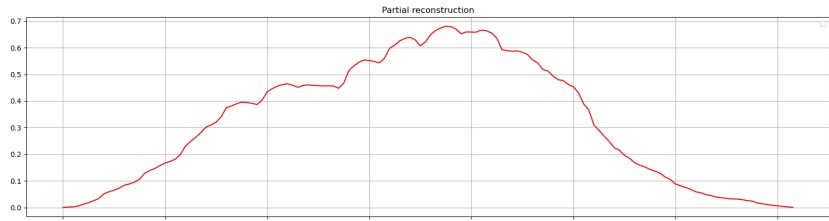

Figure 13: **Decoded unigram '1' at second position** (which is the case most of the time in our setting)

In Figure 13, we see that the pattern favorably characterizing the 'warm season' class shows a low power consumption during the middle of the day.

## A.6 ADDITIONAL RESULTS FOR TRANSFER LEARNING

For the transfer Section 4.3, the transfer allows us to greatly improve our results where we were not very efficient before. Table 8 shows how the transfer allows us to improve on these datasets.

Table 8: Results for the above datasets benefiting from the transfer and comparison with 1-NN methods. Bold accuracies stand for best accuracies.

| Datasets | Ours + LR | Ours (transfer) + LR | Random | 1-NN ED | 1-NN DTW (w=100) |
|---|---|---|---|---|---|
| MiddlePhalanxTW | 0.49 | **0.57** | 0.29 | 0.51 | 0.51 |
| MiddlePhalanxOutlineCorrect | 0.70 | **0.74** | 0.57 | 0.77 | 0.70 |
| MiddlePhalanxOutlineAgeGroup | 0.49 | **0.62** | 0.57 | 0.52 | 0.50 |
| DistalPhalanxOutlineAgeGroup | 0.68 | **0.77** | 0.47 | 0.63 | **0.77** |
| DistalPhalanxTW | 0.57 | **0.63** | 0.30 | 0.63 | 0.59 |
| DistalPhalanxOutlineCorrect | **0.75** | **0.75** | 0.58 | 0.72 | 0.72 |
| ProximalPhalanxOutlineAgeGroup | 0.83 | **0.86** | 0.49 | 0.79 | 0.80 |
| ProximalPhalanxTW | 0.76 | **0.80** | 0.35 | 0.71 | 0.76 |
| ProximalPhalanxOutlineCorrect | **0.82** | **0.82** | 0.68 | 0.81 | 0.78 |
| PhalangesOutlinesCorrect | **0.79** | **0.79** | 0.61 | 0.76 | 0.73 |

Table 9 presents transfer results for other time series of length 720.

Table 9: Transfer on time series 720 time steps long. Columns indicate the source datasets whose unsupervised architectures are re-used. Rows indicate the target datasets. Bold accuracies stand for the best results for each target dataset. Underlined accuracies stand for the result without transfer.

| Target datasets | Computers | RefriDevices | ScreenType | SmallKA | LargeKA |
|---|---|---|---|---|---|
| Computers | **0.77** | 0.63 | 0.71 | 0.60 | 0.67 |
| RefriDevices | 0.47 | 0.47 | **0.48** | 0.46 | 0.48 |
| ScreenType | **0.53** | 0.46 | 0.49 | 0.38 | 0.42 |
| SmallKA | 0.69 | 0.67 | 0.61 | 0.70 | **0.71** |
| LargeKA | **0.78** | 0.73 | 0.71 | 0.67 | 0.77 |

## A.7 UNSUPERVISED ARCHITECTURE VISUALIZATION

Figure 14: **Unsupervised architecture details**

## A.8 COMPUTE RECEPTIVES FIELDS REGIONS

With our model, it is very useful to be able to calculate the receptive fields and in particular the receptive field regions relative to an element (or a region) of the representation. The formulas have been computed by Araujo et al. (2019):

$$v_0 = v_L \prod_{i=1}^{L} s_i - \sum_{l=1}^{L} (1 + p_l - k_l) \prod_{i=1}^{l-1} s_i$$

$$u_0 = u_L \prod_{i=1}^{L} s_i - \sum_{l=1}^{L} p_l \prod_{i=1}^{l-1} s_i$$

Where :

- $v_0$ stands for the left-most coordinates of the receptive field in the intial time series
- $u_0$ stands for the right-most coordinates of the receptive field in the intial time series
- $v_L$ stands for the left-most coordinates in the representation
- $u_L$ stands for the left-right coordinates in the representation
- $k_l$ stands for the kernel size at depth l
- $s_l$ stands for the stride at depth l
- $p_l$ stands for the padding at depth l
- L stands for the depth of the network

