# OpenReview forum: "Learning Interpretable Neural Discrete Representation for Time Series Classification"
_ICLR.cc/2023/Conference — Submitted to ICLR 2023_

### Official Review · Reviewer_znQo · 2022-10-13

**Confidence:** 4
**Correctness:** 4
**Technical Novelty And Significance:** 2
**Empirical Novelty And Significance:** 2
**Recommendation:** 3

**Clarity, Quality, Novelty And Reproducibility:**

The method is rather clearly described, and Figure 1 helps to grasp the different parts of the model.
In terms of reproducibility, please refer to the discussion above on my concerns regarding the experiments.

**Strength And Weaknesses:**

* Strengths:
    * The paper deals with an important topic (interpretability in Time Series Classification)
    * The overall idea to rely on unsupervised feature extraction to improve on transferability is interesting
* Weaknesses
    * The presentation of the state-of-the-art misses important work on the topic of interpretable time series classification
    * The conducted experiments are not sufficient to convince the reader that the method is a good pick for transfer learning or interpretable TSC

These items are discussed in more details in the following.

1. Literature review

The overall claim of the paper is that the proposed model leads to improved performance on interpretability and transfer learning.
In order to support this claim, a more thorough study of the literature on the topics would be necessary, in order to position the proposition wrt previous work.

* One could cite:
    * on interpretable TSC:

        1. Fang et al. Efficient Learning Interpretable Shapelets for Accurate Time Series Classification (ICDE 2018)
        2. Wang et al. Adversarial Regularization for Explainable-by-Design Time Series Classification (ICTAI 2020)

    * on transfer learning for time series:

        3. Wilson et al. Multi-Source Deep Domain Adaptation with Weak Supervision for Time-Series Sensor Data (KDD 2020)

And it would be helpful to use these works as baselines whenever possible on the interpretability / transferability experiments to showcase the potential superiority of the proposed method.

2. Experiments

    1. Evaluation of interpretability
        * Here, interpretability is evaluated on a single example with no baseline, which makes it hard to evaluate the performance of the method in general.

    2. Problems with the experimental setting:

        > the unsupervised architecture is trained for the different possible time-scale reductions on the whole dataset (p.6)

        * This will result in unsupervised representations that might overfit the test data, hence impair the overall evaluation

        * Also, I do not understand why the related hyper-parameter $K$ is not discussed in Section 4.1.1, as it seems to be an important hyper-parameter for the method. In Figure 3, it seems that $K$ is chosen smaller for the top-scale codebook, why is that? (or did I misunderstand something?)

Some additional minor comments:

* Eq3 : the argmin should be over $\mathbf{w}, b$ I guess
* The following text appears twice in the same paragraph

    > Table 2 shows the transfer accuracy for datasets where the time series are composed of 80-time steps. For more results, see Appendix A.6
* In Section 4.3, by "codebook length", I guess the authors mean "codeword length". "codebook length" is misleading since it could correspond (or, at least, in my mind it does) to the number of atoms (codewords) in the codebook

**Summary Of The Paper:**

This paper presents a model for Time Series Classification (TSC).
The model is made of two parts: first an encoder is trained to extract interpretable features using a VQ-VAE strategy and second, a classification head builds n-grams (in practice unigrams and bigrams) and performs regularized logistic regression on the histograms of n-grams.

The model is evaluated in terms of accuracy, transferability and interpretability.

**Summary Of The Review:**

Though this paper tackles an interesting problem, my opinion is that the model is not especially novel (made of two standard blocks).
This would not be a problem if the experiments proved the superiority of the method wrt baselines but there are limitations in the conducted experiments that make the overall contribution too narrow.

---

> ### Author Response · Authors · 2022-11-14
> **Answers to the third review.**
>
> The authors would like to thank the reviewer for all the comments. Below are answers and clarifications that we hope would address the reviewer's concerns.
>
> **Q.1**: The presentation of the state-of-the-art misses ...
>
> **A.1**: Thank you for your recommendations. We will read these articles carefully. Indeed, we will prove the interest of our method compared to other interpretable methods.
>
>
> **Q.2**: The conducted experiments are not sufficient to convince the reader...
> - **Q.2.1.** : Evaluation of interpretability ...
> - **Q.2.2.**: Problems with the experimental setting ...
>
>
>  **A.2.1**: The authors agree with the reviewer.
>
>  **A.2.2**: As you mentioned, several unsupervised representations are learned for the whole dataset.
> In our opinion, it does not raise evaluation issues. The main point is to not use the test labels at any moment in the training procedure. However, we can consider that when we discover the data $X_{test}$ (the test time series), we can always re-train a representation on the concatenation of $X_{train}$ and $X_{test}$.
>
> We are not sure to understand what hyperparameter you are talking about. Does K mean the downscaling for each level of representation? This hyperparameter for the two levels of representation is the only one in the architecture we are optimizing. As described in section 4.1. they are found by grid-search by performing a classification on the validation dataset that was not used for training. Or are you talking about the K in the Franceschi et al. model we are comparing to?
>
>
> **Q.3 & Q.4**: Eq3 : the argmin should be over w,b ... The following text appears twice in the same paragraph.
>
> **A.3 & A.5**: Thank you for your feedback, indeed they are mistakes.
>
> **Q.5**: In Section 4.3, by "codebook length" ...
>
> **A.5**: To clarify the vocabulary specific to the representation.
>
> - The codebook's length corresponds to the length of the sequence of centroids. For instance, let's consider a time series of length 512 and a model with only one level of representation. Let's imagine that our encoder induces a contraction of the dimension of $2^{3}$. The codebook's length will then be 64.
>
> - The support of a fixed-time element in the representation is equal to the number of centroids available during the vector quantization. As stated in section 4.1, we set at 32 for each level of representation.

---

> > ### Comment · Reviewer_znQo · 2022-11-15
> > **Response to the authors**
> >
> > Thank you for this detailed and honest answer.
> >
> > Regarding the question about $K$: I was mistaken, please ignore my comment on this point.
> >
> > Overall, given your answer, I will stick to my initial rating.
> >
> > Best regards

---

### Official Review · Reviewer_icKJ · 2022-10-24

**Confidence:** 4
**Correctness:** 2
**Technical Novelty And Significance:** 2
**Empirical Novelty And Significance:** 2
**Recommendation:** 5

**Clarity, Quality, Novelty And Reproducibility:**

The paper is clearly written, with enough details and code to reproduce. The idea is novel in the time series classification domain.

**Strength And Weaknesses:**

Pros:
1. The interpretability in time series feature extraction, is largely of interests to the community.
2. The paper is well written and easy to follow. The description of the complete system including the probing logistic regression task, is clear to understand. The Figure 1 helps a lot.
3. The details of practical tricks including stop-gradient, the regularizer to stabilize training, etc, are provided.

Cons:
1. The property "we intend two identical elements of the representation to be decoded as the same pattern regardless of their position", is not equivariance, but invariance. Equivariance requires a transformation in the input leads to a similar transformation in the output, e.g. x -> f(x), x+d -> f(x)+f(d). What described here is time-invariance or stationary process.
2. The interpretability claim is very handwaving. From the example showed in Figure 4, one can easily detect the peak value difference from the original time series (left two figures). And shapes are also similar in the original series. The feature only verified this. To provide a more conniving example, the author should use some examples that is very hard to differentiate from the original time series, but easy to detect in the feature space, and the features (at least part of them) have a clear physical meaning for human to understand.
3. The final performance is not as good as other TSC baselines.
4. The transfer learning task does not have baselines to compare, thus hard to judge the value.


**Summary Of The Paper:**

This paper proposes an interpretable representation learning method to extract features from time series and use them for the classification task. The authors borrow the idea from unsupervised autoencoder, vector quantization, multi-level encoding and interpretable probing tasks. Joint learning with decoding recovery loss plus losses associated with the codebook, lead to a feasible and stable training framework for representation learning in time series. The author claims such scheme offers interpretable features from encoding, and can be used in time series classification and transfer learning tasks.

**Summary Of The Review:**

Overall, given the fact that the proposed method is not competitive to other existing methods, and the important claim of interpretability is handwaving and vague, I feel there is lots of rooms to improve this paper.

---

> ### Author Response · Authors · 2022-11-14
> **Answers to the second review.**
>
> The authors would like to thank the reviewer for all the comments. Below are answers and clarifications that we hope would address the reviewer's concerns.
>
>
> **Q.1**: The property "we intend two identical elements ...
>
> **A.1**: We probably poorly framed this sentence. We indicated that an element of the representation at a given position and the same element of the representation at a different position are reconstructed in the same way with a shift. Let's consider the codebook $(0 \ 0 \ \boldsymbol{a} \ 0  \ 0)$ and the codebook $(0 \ 0 \ 0 \ 0 \ \boldsymbol{a})$, centroid **a** is decoded into the same values in the initial time domain with a shift. See Appendix A.2. for a formal definition of equivariance.
>
>
> **Q.2**: The interpretability claim is very handwaving...
>
> **A.2**: Thank you for your comment. Indeed, the use case treated in section 4.2 shows that the solution to the classification problem relies on the power peak level of the two devices. So it is mainly a level-difference difference. As you mentioned, this is visible in the original space of the time series to a trained observer. In appendix A.5. we dealt with another use case. This interpretable use case shows us which frequency level is relevant for the classification problem and allows us to visualize some particularly discriminating features. In this case, the differences are not level-based but shape-based.
>
>
> **Q.3**: The final performance is not as good as other TSC baselines.
>
> **A.3**: Compared to recent supervised end-to-end methods, our average accuracy is lower. However, these models are not interpretable at the model level. When comparing to unsupervised models, you can see that the performance of our model is quite good.
> We agree with the reviewer that we could have compared with other interpretable models for TSC.
>
> Nevertheless, we did not try to get the best possible accuracy from the logistic regression on the representations. Our goal was to exhibit that our model which provided interpretability has honorable classification performances with minimal intervention (simple logistic regression on unigrams/bigrams) on the representation.
>
> **Q.4**: The transfer learning task does not...
>
> **A.4**: Indeed, we do not compare our transfer performances to other transfer methods. It is quite difficult to find metrics to compare the transfer of our unsupervised model to other transfer methods.
>
> **Summary Of The Review**
>
> **Q.5**: Overall, given the fact that the proposed method is not competitive to other existing methods, and the important claim of interpretability is handwaving and vague, I feel there is lots of rooms to improve this paper.
>
> **A.5**: Thank you for your feedback and for the recommended progress areas.

---

> > ### Comment · Reviewer_icKJ · 2022-11-16
> > **Thanks for your answers**
> >
> > Thanks for the responses to my concerns. I feel with proper rewriting and more interpretability analysis, this would become a good paper in the field. I keep my original ratings.

---

### Official Review · Reviewer_5jZC · 2022-10-30

**Confidence:** 4
**Correctness:** 2
**Technical Novelty And Significance:** 2
**Empirical Novelty And Significance:** 2
**Recommendation:** 3

**Clarity, Quality, Novelty And Reproducibility:**

The overall approach has promise but there are too many unanswered questions and parts that are either poorly explained or lack clarity or adequate comparisons. I also dont understand what makes the proposed approach particularly novel.

**Strength And Weaknesses:**

The authors try to tackle an important problem and provide some examples of how it works in multiple experimental domains.

The paper has several weaknesses.

1) The authors never make it clear what they mean by interpretability in this particular context. Yet interpretability is inherently tied to the downstream task so its difficult to understand what any guarantees really mean here. Do the authors mean sparsity? Are sparse models all interpretable? What happens if the features themselves are not interpretable eg pixels in an image?

2) The authors are missing several important contributions to understanding timeseries data. For example, Wu et al 2022 (see thesis https://dash.harvard.edu/handle/1/37371748) present a way of understanding and summarizing the crucial aspects of a time series using concepts. How does the proposed approach of learning a discrete dictionary compare to the idea of using concept bottleneck models to summarize important parts of the time series

3) The biggest weakness is the lack of convincing application in the experiments section. The authors use the UCR dataset and explicitly state that validation is an issue because of lack of sample size so they instead focus on the sets of reasonable size yet not many details are provided about this. How long are the time series? How does performance vary over varying length of time series? If you learn a poor yet interpretable representation of the time series at a particular point, does it compound throughout the time series?

4) The authors talk about the accuracy vs interpretability tradeoffs. I would like the authors to rethink that statement in light of this fantastic paper https://www.nature.com/articles/s42256-019-0048-x

5) It is unclear how the approach generalizes across different applications and modalities of data.

**Summary Of The Paper:**

The authors present a neural network based approach for managing the so called tradeoff between interpretability and accuracy in timeseries data by learning a dictionary of discrete representations. They guarantee that 1) only a small number of patterns that can be visualised easily are learnt 2) training a linear classifier over a limited number of patterns provides a more explainable decision 3) a shift equivalence property of the model that is associated with a time consistency of the representation.

**Summary Of The Review:**

The overall approach has promise but there are too many unanswered questions and parts that are either poorly explained or lack clarity or adequate comparisons. For this reason I am rejecting the paper.

---

> ### Author Response · Authors · 2022-11-14
> **Answers to the first review.**
>
> The authors would like to thank the reviewer for all the comments. Below are answers and clarifications that we hope would address the reviewer's concerns.
>
> **Q.1:** The authors never make it clear what they mean by...
>
> **A.1:** In section 2 of the paper, we have presented the different types of interpretability (instance or class-wise) with relevant literature and have clearly stated in which kind our model position itself: class-wise interpretation based on a neural representation of patterns.
> The interpretability then defined (representation of patterns) does not involve sparsity during classification but is provided by the construction of unsupervised representation. The sparsity in this paper is provided through the $\ell_1$ penalty and is a well-known practice when building logistic regression models and ease visualization.
>
>
> **Q.2:** The authors are missing several important contributions...
>
> **A.2:** Thank you for the recommendation. The work referred to is a recent bachelor thesis based on this paper:  Koh, P.W., Nguyen, T., Tang, Y.S., Mussmann, S., Pierson, E., Kim, B. &amp; Liang, P.. (2020). Concept Bottleneck Models. Proceedings of the 37th International Conference on Machine Learning.
> The original bottleneck models have been introduced for image representation (knee x-ray grading and bird identification). To our knowledge, there is no published paper that applied this concept to time series data.
> Moreover, the bottleneck model defined in the reference provided is a totally different concept from what we propose here:
> - our interpretability does not come from spare feature selection (it does in the bottleneck reference)
> - our model is not an optimization method to enhance the selection of existing machine learning (through a supplementary layer insertion) but provides an unsupervised representation of the time series which is interpretable (please see appendix A.2. to better understand the pattern recognition and the equivariance principle).
>
> **Q.3 & Q.5**: The biggest weakness is the lack of convincing ...
>
> **A.3 & A.5**: The use case chosen in the core of the article is perhaps insufficient to convince of the interest in the method in terms of interpretability. Maybe the use case in the appendix is more convincing.
> Regarding the choice of the UCR datasets, they are very common datasets (from various applications like electrical signals from devices, sensors, images, EEG, etc.) used in TSC. It helps us demonstrate that the classification performances obtained by our model are comparable to the best-unsupervised methods in the literature when we provide interpretability which other models do not. Moreover, we explain in section 4.1 that we perform experiments on 39 datasets of various natures (electrical signals from devices, sensors, images, EEG, etc.). We also present the results by dataset type in Table 1.
>
>
> **Q.4**: The authors talk about the accuracy vs interpretability tradeoffs...
>
> **A.4**: Our goal here is to propose an interpretable model with good classification performance. We think that our model is not a black box as we designed it to understand how the decision process is made (one more time please refer to appendix A.2. for a better visual understanding).

---

### Decision · Program_Chairs · 2023-01-20

**Decision:**

Reject

**Justification For Why Not Higher Score:**

The paper makes a number of sound claims, most of them not substantiated sufficiently. The interpretability is a central contribution claimed by the authors, but the paper lacks a clear definition of 'interpretability'. The paper is not clearly written and the empirical evaluations are not adequate.

**Justification For Why Not Lower Score:**

N/A

**Metareview: Summary, Strengths And Weaknesses:**

The paper presents an interpretable and accurate neural convolutional architecture for time-series data sets. Although the authors make a number of claims (learning small number of patterns allows for interpretability through visualization, time-consistent representations etc), there is little evidence to substantiate these claims. The term 'interpretability' has been very loosely used and the paper is unclear in many parts. Empirical evaluations are not sound too.